# Membrane Properties of Human Induced Pluripotent Stem Cell-Derived Cultured Red Blood Cells

**DOI:** 10.3390/cells11162473

**Published:** 2022-08-09

**Authors:** Claudia Bernecker, Eva Maria Matzhold, Dagmar Kolb, Afrim Avdili, Lisa Rohrhofer, Annika Lampl, Martin Trötzmüller, Heike Singer, Johannes Oldenburg, Peter Schlenke, Isabel Dorn

**Affiliations:** 1Department of Blood Group Serology and Transfusion Medicine, Medical University of Graz, 8036 Graz, Austria; 2Core Facility Ultrastructure Analysis, Medical University of Graz, 8010 Graz, Austria; 3Gottfried Schatz Research Center for Cell Signaling, Metabolism and Aging, Division of Cell Biology, Histology and Embryology, Medical University of Graz, 8010 Graz, Austria; 4Core Facility Mass Spectrometry, Center for Medical Research, Medical University of Graz, 8010 Graz, Austria; 5Institute of Experimental Haematology and Transfusion Medicine, University Clinic Bonn, 53127 Bonn, Germany

**Keywords:** red blood cells, induced pluripotent stem cells, erythropoiesis, membrane, blood group antigen, deformability, osmotic resistance, phospholipids, cholesterol, reticulocytes

## Abstract

Cultured red blood cells from human induced pluripotent stem cells (cRBC_iPSCs) are a promising source for future concepts in transfusion medicine. Before cRBC_iPSCs will have entrance into clinical or laboratory use, their functional properties and safety have to be carefully validated. Due to the limitations of established culture systems, such studies are still missing. Improved erythropoiesis in a recently established culture system, closer simulating the physiological niche, enabled us to conduct functional characterization of enucleated cRBC_iPSCs with a focus on membrane properties. Morphology and maturation stage of cRBC_iPSCs were closer to native reticulocytes (nRETs) than to native red blood cells (nRBCs). Whereas osmotic resistance of cRBC_iPSCs was similar to nRETs, their deformability was slightly impaired. Since no obvious alterations in membrane morphology, lipid composition, and major membrane associated protein patterns were observed, reduced deformability might be caused by a more primitive nature of cRBC_iPSCs comparable to human embryonic- or fetal liver erythropoiesis. Blood group phenotyping of cRBC_iPSCs further confirmed the potency of cRBC_iPSCs as a prospective device in pre-transfusional routine diagnostics. Therefore, RBC membrane analyses obtained in this study underscore the overall prospects of cRBC_iPSCs for their future application in the field of transfusion medicine.

## 1. Introduction

In modern medicine, the transfusion of red blood cells (RBCs) is essential in treating numerous diseases. RBC units are mainly derived from allogeneic donations of healthy volunteers. Already today, shortages exist in blood supply for patients with rare blood group phenotypes or those suffering from severe alloimmunization. Worsening of the overall situation within the next decades due to demographical changes is expected [1]. A promising option for reducing shortages in allogeneic RBC supply is the ex vivo manufacturing of cultured RBC (cRBCs) from human stem cells. Due to their self-renewing properties and their ability to differentiate into all tissues of the human body, induced pluripotent stem cells (iPSCs) offer a very attractive stem cell source for mandatory large-scale production of cRBCs [2,3]. It was estimated that iPSC lines of a few donors would be adequate to cover the requirements of most alloimmunized patients and those with rare blood group phenotypes [4]. Reprogramming of patient-derived cells into iPSCs would even allow for the transfusion of autologous cRBCs in individual cases. Besides their use as therapeutics, iPSC-derived cRBCs (cRBC_iPSCs) could offer an attractive diagnostic tool in terms of RBC test panels. Laboratories use RBC panels for compulsory screening and identification of irregular RBC antibodies before transfusion. The continuous production of RBC test panels is challenging and depends on the accessibility of blood group O donors, having the relevant blood group antigens.

Since the pioneering discovery that somatic cells can be reprogrammed for pluripotency, several culture systems for the ex vivo generation of cRBC_iPSCs have been established [5,6]. Recent approaches already focus on the transfer of static cell culture methods into scalable bioreactor systems [7,8]. However, the safety and functionality of cRBC_iPSCs must be thoroughly shown before these cells can be used in clinical or laboratory settings. Thus far, the low efficiency of established culture systems, in terms of expansion and enucleation, have prevented sufficient generation of enucleated cRBC_iPSCs for functional analyses [2,3,5]. Especially, the absence of a physiological niche might impair cell growth and lineage-specific differentiation. Our group recently reported improved erythroid differentiation of iPSCs by providing cellular interactions in a three-dimensional structure termed “hematopoietic cell forming complex” (HCFC) [9]. From this HCFC, CD43+ hematopoietic cells (purity > 95%) were released into the culture supernatant over a period of 3–5 weeks and could be collected repeatedly for further differentiation into glycophorin A (GPA) positive erythroid cells. Mean enucleation rates near 40% (up to 70%) allowed for the production of several millions of enucleated cRBC_iPSCs from one six-well plate. The final output and overall quality of cultured cells was further improved by optimized cholesterol supplementation during culturing [10]. In the present study, we confirmed the efficacy of our established system and conducted a comprehensive functional characterization of enucleated cRBC_iPSCs with a focus on membrane properties. As illustrated in Figure 1, cRBC_iPSCs were analyzed for their maturation stage, morphology, deformability, osmotic resistance, lipid profile, and expression of transfusion relevant blood group antigens and compared with peripheral blood (PB)-derived native RBCs (nRBCs) and cord blood (CB)-derived native reticulocytes (nRETs). These controls differ not only in their maturation stage, but they also represent different developmental waves of human erythropoiesis. Whereas CB cells originate from fetal liver erythropoiesis, PB cells originate from adult bone marrow (BM) erythropoiesis. There is increasing evidence that RBCs from different developmental waves differ in their cellular features like hemoglobin (Hb) composition, Hb content, and cell size [10,11,12].

## 2. Materials and Methods

### 2.1. Human Material

Five human iPSC lines derived from erythroblasts or CB CD34+ cells were used [9,13,14,15] (Appendix A). Native RBCs were obtained from RBC units within 24 h after donation, and nRETs were isolated from CB with magnetic beads within 12 h postpartum (CD71 Microbead Kit; Miltenyi Biotec, Bergisch Gladbach, Germany). Written informed consent was given prior to sampling. The study was approved by the local ethics committee in line with the Declaration of Helsinki (EK 27 165ex 14/15).

### 2.2. Cultivation of cRBC_iPSCs

Hematopoietic and erythroid differentiation of iPSCs were induced as recently described [9], and illustrated in Figure 1. For embryoid body (EB) formation, colonies (>23 passages) were detached with collagenase IV (Sigma-Aldrich, St. Louis, MO, USA). Cell clumps were seeded on low-binding suspension plates (Nunclon Sphera, Thermo Fisher Scientific, Waltham, MA, USA) and incubated for 5 days in hESC medium without bFGF [13]. Afterward, spherical EBs were transferred into six-well tissue culture plates (Sarstedt, Nümbrecht, Germany) with STEMdiff APEL2 Medium (STEMCELL Technologies, Vancouver, BC, Canada), 5% Protein-Free Hybridoma Medium (Thermo Fisher Scientific, Waltham, MA, USA), 5 ng/mL interleukin-3 (IL-3; PeproTech), 100 ng/mL SCF (PeproTech, London, UK), and 3 U/mL erythropoietin (EPO) (Erypo, Janssen Biologics BV, Leiden, Netherlands). The medium was replaced weekly. For erythroid differentiation, single cells released into the supernatant were collected and cultured for an additional 18 days in an established three-phase erythropoiesis assay [9,10]. Cells were cultured in Iscove’s medium (Biochrom, Berlin, Germany) with 5% human plasma (Octapharma, Vienna, Austria), 10 μg/mL insulin (Sigma-Aldrich, St. Louis, MO, USA), 330 μg/mL human holotransferrin (BBI Solutions, Salisbury, UK), and from day 8 onward 4 mg/dl cholesterol-rich lipids (Sigma-Aldrich, St. Louis, MO, USA). Cells were stimulated with 100 ng/mL SCF, 5 ng/mL IL-3, 3 U/mL EPO, and 10^−6^ M hydrocortisone (Sigma-Aldrich, St. Louis, MO, USA) from days 0 to 8, with 100 ng/mL SCF and 3 U/mL EPO from day 8 to day 11, and with 3 U/mL EPO from days 11 to 18. Cells were filtered through a syringe filter (Acrodisc WBC Pall, Port Washington, NY, USA), to obtain the pure enucleated portion.

### 2.3. Phenotype and Maturation Stage of cRBC_iPSCs

Phenotypic characterization was conducted via microscopic assessment of cytospin preparations and via flow cytometry. For flow cytometry analyses, cells were incubated with respective antibodies against CD45, CD71 (Becton Dickinson, Franklin Lakes, NJ, USA), CD36, CD49d, GPA (Beckman Coulter, Krefeld, Germany), and band 3 (Bric 6, IBGRL, Bristol, UK). Measurements were conducted using a CytoFLEX flow cytometer (Beckman Coulter, Krefeld, Germany). The maturation stage of filtered cells was evaluated on the basis of thiazole orange staining (Retic Count, Becton Dickinson) and the expression of CD71. Cells were further characterized after staining with new methylene blue (Reticulocyte Stain, Sigma-Aldrich), and at least 300 cells were enumerated microscopically (Axioscope, Zeiss, Oberkochen, Germany). Cell size was assessed via flow cytometry using scatter characteristics and via microscopy (EVOS M5000, Thermo Fisher Scientific, Waltham, MA, USA) after staining with May–Gruenwald–Giemsa (Hemafix, Biomed, Oberschleißheim, Germany). The hemoglobin (Hb) concentration was measured using Drabkin’s Reagent (Sigma-Aldrich, St. Louis, MO, USA). A total of 7 × 10^6^ cells were incubated with 500 µL Drabkin’s solution and read at a wavelength of 540 nm on a Shimadzu-1800 spectrophotometer (Shimadzu, Korneuburg, Austria).

### 2.4. Electron Microscopy

For scanning electron microscopy (SEM), cells were processed as published [10]. For transmission electron microscopy (TEM), samples were fixed in 2% PFA/2.5% glutaraldehyde for 1 h, post-fixed in cacodylate buffer/OsO4 for 1 h, and afterwards washed in cacodylate buffer. After dehydration, samples were infiltrated (propylene oxide and TAAB embedding resin, pure TAAB embedding resin) for 3 h, placed in TAAB embedding resin (2x 90 min), transferred into embedding molds, and polymerized (72 h, 60 °C). Sections were stained with lead citrate and platinum blue (International Bio-Analytical Industries, Inc., Boca Raton, FL, USA), and investigated at 120 kV with a Tecnai G 2 FEI microscope (FEI, Eindhoven, Netherlands) equipped with a Gatan ultrascan 1000 CCD camera.

### 2.5. Osmotic Resistance

Cells were incubated in decreasing concentrations of NaCl (0.9–0%), and the amount of free Hb in the supernatant was measured using a UV-1800 spectrophotometer (Shimadzu, Korneuburg, Austria), as described [10,16].

### 2.6. Deformability

Cells were examined on a laser optical rotational cell analyzer (Lorrca MaxSis; RR Mechatronics Hoorn, AN Zwaag, Netherlands) according to an established protocol [17].

### 2.7. Immunofluorescence

After fixation with 1% paraformaldehyde (Santa Cruz Biotechnologies, Dallas, TX, USA) and permeabilization with 0.05% saponin (Sigma-Aldrich, St. Louis, MO, USA), cells were incubated with anti-alpha 1 spectrin antibodies (abs) (mouse monoclonal); anti-ankyrin erythroid/ANK abs (mouse monoclonal); anti-band 3/AE 1 abs (rabbit monoclonal) or f-actin staining kit, respectively. Secondary abs were Alexa Fluor 488 goat anti-mouse IgG or Alexa Fluor 555 donkey anti-rabbit IgG (all Abcam, Cambridge, UK). Unspecific reactions were blocked with 10% goat serum (Sigma-Aldrich, St. Louis, MO, USA). Imaging was performed on a Nikon Eclipse Ti microscope (Amstelveen, the Netherlands). The mean fluorescence intensity over the whole image field was quantified with the Nikon General Analysis 3 Software (version 1.3.660.548).

### 2.8. Lipidomics Analysis

The quantitative evaluation of cholesterol (chol) and the phospholipids phosphatidylcholine (PC), phosphatidylserine (PS), sphingomyelin (SM), phosphatidylethanolamine (PE), phosphatidylinositol (PI), and lysophosphatidylcholine (LPC) was conducted using high-resolution mass spectrometry, as recently described by our group [10]. Lipids were extracted from cell pellets (10^7^ cells) with an established MTBE protocol [18]. Data acquisition was performed using Orbitrap-MS (Orbitrap Velos Pro hybrid mass spectrometer, Thermo Fisher Scientific, Waltham, MA, USA) [19]. Complete scan profile spectra from m/z 320 to 1050 for positive ion mode and from m/z 500 to 1000 for negative ion mode were obtained in the Orbitrap mass analyzer at a resolution of 100k at m/z 400 and <2 ppm mass accuracy. All samples were measured once in positive polarity and once in negative polarity. For MS/MS experiments, the 10 most abundant ions of the entire scan spectrum were sequentially fragmented in the ion trap using He as collision gas (normalized collision energy: 50; isolation width: 1.5; activation Q: 0.2; and activation time: 10), and centroided product spectra at a normal scan rate (33 kDa/s) were obtained. Data analysis was accomplished by Lipid Data Analyzer, a custom-developed software tool [19,20,21].

### 2.9. Blood Group Phenotyping

Expression of blood group antigens of the systems ABO, Rhesus (D, C, c, E, e, Cw), Kell (K, k, Kp), Kidd (Jk), Duffy (Fy), and MNS (M, N, S, s) were analyzed with commercially available test systems (Appendix A) used in our routine patient diagnostic according to the manufacturer guidelines (all from BioRad, Vienna, Austria). For each antigen, 1–2 × 10^6^ washed RBCs were diluted with 12.5 µL Diluent 1 or Diluent 2, incubated for 10 min if specified by the manufacturer, and transferred into respective gel cards having equivalent antibodies. For detecting Duffy antigens, RBC suspensions were incubated with 25 µL Diluent 2 and 12.5 µL anti-Fya or anti-Fyb for 15 min at 37 °C before centrifugation. To detect antigens S or s, cells were incubated with 25 µL Diluent 2 and 12.5 µL anti-S or anti-s before transfer into LISS/Coombs ID-cards. After 10 min centrifugation of gel cards in a DiaMed ID-Centrifuge 24 S, results were obtained visually.

### 2.10. Blood Group Genotyping

Genomic DNA was prepared with the Qiamp DNA Mini Kit (Qiagen, Hilden, Germany). Genotyping was conducted using the ABO Type variant Kit (BAGene, BAG Health Care, Lich, Germany), the RBC-Ready Gene kits CDE, Zygofast, MNS, and KKD, and the Ready Gene Rare ID Kit (all from Innotrain Diagnostik, Kronberg, Germany).

### 2.11. Statistical Analysis

Statistical analyses were performed with IBM SPSS statistics 27. The data were analyzed using the non-parametric Mann–Whitney U test and the Kruskal–Wallis test with subsequent Bonferroni correction. Values of *p* < 0.05 were considered statistically significant.

## 3. Results

### 3.1. Generation of cRBC_iPSCs

Hematopoietic and erythroid differentiation was induced from five well-characterized human iPSC lines (iPSC 1-5; Appendix A) using an established cell culture system, as visualized in Figure 1 [9,10]. Cultured RBC_iPSCs (>99% GPA+ and band 3+, Appendix A) were filtered to obtain the pure enucleated RBC portion for analyses of membrane features. As evaluated by microscopy, the purity of enucleated cells was 94.7% ± 5.2% (n = 21). For each of the following analyses, cRBCs from at least three different iPSC lines were examined and compared with their native equivalents, nRBCs and nRETs.

### 3.2. Maturation Stage of cRBC_iPSCs, Cell Size, and Hemoglobin Content

As shown by new methylene blue staining for the remaining reticulofilamentous material, nRETs indicated a homogeneous maturation stage of young reticulocytes with dense, dark-blue chromatin residues, whereas chromatin residues were absent in nRBCs. cRBC_iPSCs indicated some remaining chromatin spots and therefore a maturation stage between nRETs and nRBCs on the cytoplasmic level (Figure 2A, Appendix A). This was confirmed via flow cytometry using thiazole orange staining (Figure 2B,C, Appendix A). Besides the degradation of residual RNA and intracellular organelles like mitochondria and ribosomes [22,23], terminal maturation is associated with loss of the transferrin receptor CD71, the thrombospondin receptor CD36, and α4-integrin (CD49d) [24,25]. Although the expression of CD36 (21.5% ± 12.3%) and CD49d (2.9% ± 2.1%) was low in cRBC_iPSCs (Appendix A), the expression of CD71 remained high (94.4% ± 3.3%).

Cell diameter assessed using microscopy was higher for cRBC_iPSCs (11.2 ± 1.5 µm) than for nRETs (8.1 ± 2.2 µm, *p* < 0.001) and nRBCs (7.4 ± 0.8 µm, *p* < 0.001) (Figure 3A, Appendix A). Additionally, cell size was determined based on scatter characteristics derived from flow cytometry. The mean fluorescence intensity (MFI) in the forward scatter was slightly higher in cRBC_iPSCs than in nRETs, without attaining statistical significance. Both cell types displayed a significantly higher MFI in the forward scatter compared with nRBCs (Figure 3B,C). This indicates a comparable cell size between cRBC_iPSCs and nRETs but higher than nRBCs. Side scatter features were also comparable between cRBC_iPSCs and nRETs, indicating similar granularity due to the residual intracellular material of young enucleated cells. Both sources varied significantly from mature nRBCs, devoid of RNA and intracellular organelles (Figure 3B,D). Photometric evaluation of Hb content using Drabkin’s solution indicated an Hb concentration in cRBC_iPSCs (205.6 ± 46 mg/mL) comparable with that in CB-derived nRETs (201.4 ± 29.7 mg/mL), but higher than in PB-derived nRBCs (131.3 ± 17.9 mg/mL). All measurements were performed with comparable cell concentrations (10^7^ cells in a total volume of 500 µL Drabkin’s solution).

### 3.3. Biomechanical Characterization of cRBC_iPSCs

We evaluated the main biophysical membrane functions of RBCs, osmotic resistance (OR), and deformability. The OR was analyzed by measuring the free Hb content of RBCs in declining NaCl concentrations (Figure 4A). Investigation of nRBCs resulted in an S-shaped curve with 50% hemolysis between 0.45% and 0.4% NaCl concentration and a sharp transition point. By contrast, the analysis of nRETs showed an S-shaped curve with 50% hemolysis between 0.35% and 0.3% NaCl. Thus, nRETs revealed a higher stability against osmotic changes than nRBCs, although this was not statistically significant. Cultured RBC_iPSCs showed a similar OR to nRETs with 50% hemolysis at 0.3% NaCl, but higher OR than nRBCs in the range of 0.4% to 0.3% NaCl (*p* < 0.001).

The deformability of cRBC_iPSCs was evaluated by Lorrca analysis (Figure 4B). The Elongation Index (EI) was calculated to describe the deformation of the cells in relation to the applied shear stress. Compared with the deformability curve of nRBCs (EImax 0.6 ± 0.01), the maximum EI was lower in nRETs (0.53 ± 0.02). The EImax of the cRBC_iPSCs was lower than in these two physiological sources (0.44 ± 0.01), which reached significance only in comparison to nRBCs (*p* < 0.001). Lower deformability of cRBC_iPSCs than that of nRBCs was also seen in the lower range from 0.5 to 10 Pa, more closely simulating physiological conditions.

### 3.4. Membrane Lipid Composition of cRBC_iPSCs

Quantitative measurements for the total cellular content of cholesterol (chol) and the phospholipids PC, PS, SM, PE, PI, and LPC were conducted using high-resolution mass spectrometry. In nRBCs, chol represented the largest portion, with 54.3% ± 7.8% of the total lipid content, followed by PC (17.8% ± 4.0%) and SM (16.2% ± 2.2%). All the other lipids constituted the remaining 11.7% (Figure 5A). Similar patterns were observed in nRETs and in cRBC_iPSCs. In comparison with native cells, the relative chol content was slightly reduced in cRBC_iPSCs (45.3% ± 2.1%), reaching significance only in comparison to nRBCs (*p* < 0.01) but not to nRETs. There were no significant differences found for the entire amount (nmol/10^7^ cells) of chol and individual phospholipid classes between cRBC_iPSCs and nRETs. Slightly elevated levels of PS and PI in cRBC_iPSCs compared to nRETs did not reach statistical significance. In line with their larger cell size and membrane surface area, the total quantity of lipids was higher in nRETs (15.9 ± 4.2 nmol/10^7^ cells) and cRBC_iPSCs (16.9 ± 0.9 nmol/10^7^ cells) than in nRBCs (7.6 ± 1.3 nmol/10^7^ cells), (*p* < 0.05). Higher absolute values were measured for chol and the phospholipids PE, SM, PC, PS, and PI (Figure 5B). LPC was the only lipid that was more abundant in nRBCs than in nRETs (*p* < 0.001) and cRBC_iPSCs (*p* < 0.05).

### 3.5. Cytoskeletal and Membrane Proteins

The expression and arrangement of ankyrin, α-spectrin, and f-actin, and the most abundant membrane protein, band 3, were envisaged by immunofluorescence (IF) analyses (Figure 6). The fluorescence patterns of ankyrin, spectrin, and band 3 did not obviously differ between cRBC_iPSCs, nRETs, and nRBCs. Although ankyrin seemed to be slightly lower expressed in nRBCs, the MFI over the whole of the image fields did not differ significantly between cRBC_iPSC, nRETs, and nRBCs (Appendix A). In nRETs and cRBC_iPSCs, f-actin showed characteristic spots of actin-agglomeration, which could be residues of the contractile actin ring after enucleation in these more immature cell categories. By contrast, terminally differentiated nRBCs were lacking these spots.

### 3.6. Morphology of cRBC_iPSCs

We conducted analyses of cell structure via scanning electron microscopy (SEM). As represented in Figure 7A, cRBC_iPSCs indicated similar morphology to multilobular nRETs or even a proto-biconcave shape. Neither echinocyte formation nor other apparent morphological changes were observed. Interestingly, nRETs showed “pit-like” structures in their membrane, and in cRBC_iPSCs, these pits were more abundant and appeared to have a higher diameter. We also noticed bubble-like excesses comparable with exosomes on the cell surface. Transmission electron microscopy (TEM) (Figure 7B) revealed a homogeneous cytosolic structure of cRBC_iPSCs and nRETs and the occurrence of some large intracellular vesicles comparable with multivesicular endosomes (MVE).

### 3.7. Blood Group Antigen Expression

Cultured RBCs from three iPSC lines were analyzed for the expression of most clinically relevant blood group antigens. Phenotyping was performed in a gel-card based system as used in our routine patient diagnostic (Figure 8 and Appendix A). Blood group antigens of the corresponding, undifferentiated iPSCs were verified by genotyping. As summarized in Table 1 for cRBCs from all three iPSC lines, expression of antigens of the systems ABO, Rhesus (D, C, c, E, e, Cw), Kell (K, k, Kp), Kidd (Jk), and MNS (M, N, S, s) were similar to the genotype of the iPSCs. The only inconsistency was noticed for the *Fyb* allele of the Duffy antigen system. Although the expression of Fya was similar to the iPSC genotype, we were unable to verify the surface expression of the Fyb blood group antigen in cRBC_iPSCs, independent of its homogeneous or heterozygous expression.

## 4. Discussion

Our study investigated the membrane properties of cRBC_iPSCs as the prerequisite for their prospective diagnostical or clinical application. The RBC membrane comprises a lipid bilayer with embedded proteins and is anchored to an elastic network of skeletal proteins [26]. The membrane architecture determines the antigenic properties as well as the mechanical features of RBCs. Altered stability or deformability of cells results in declined half-life because of splenic sequestration or even lysis and will therefore determine the survival of cRBC_iPSC after transfusion. Since cRBCs from human hematopoietic stem or progenitor cells (HSPCs) or iPSCs do not reach complete terminal maturation into biconcave shaped RBCs ex vivo [27,28], we initially determined the maturation stage and morphology of cRBC_iPSCs by their comparison with nRETs and nRBCs. Contrary to the fragile and less deformable multilobular reticulocyte, the biconcave RBC demonstrates high shear resistance and membrane flexibility in the blood flow. This permits the RBC to undergo large passive and reversible deformations during continuous capillary passages [26]. In vivo, the terminal transition of RETs to biconcave RBCs is associated with RNA degradation and organelle clearance. Concurrently, enormous membrane remodeling occurs, reducing the volume and cell surface area by 20–30% [26,29]. The mechanisms involved in terminal maturation steps are poorly defined. According to recent reports, loss of membrane surface occurs by an endosome–exosome pathway, whereas degradation of residual organelles includes autophagic mechanisms. Both pathways might interfere [22,23,30,31]. In our study, enucleated cRBC_iPSCs exhibited some residual chromatin spots and sporadic organelles, most of them already encapsulated in vacuoles for further degradation or exocytosis, and therefore a maturation stage between nRETs and nRBCs on the cytosolic level. In line with terminal membrane maturation, cell surface expression of CD36 was low, and CD49d was almost absent [25]. Nevertheless, expression of the transferrin receptor CD71 was persistent and did not correlate with other indications of terminal membrane reorganization [24,25]. This might be caused by insufficient iron supply (330 µg/mL holo-transferrin) during massive ex vivo amplification, compensated by prolonged expression of the transferrin receptor [23]. Further studies with enhanced iron supply during culturing will be necessary to confirm this hypothesis. Cell size and Hb content of cRBC _iPSCs were enhanced and more comparable with those of nRETs than nRBCs. As discussed in more detail below, cell size and Hb content might not only be influenced by the maturation stage of erythroid cells, but also by the developmental wave of human erythropoiesis [12]. Besides different growth kinetics and altered expression of transcription factors during erythropoiesis [32,33], fetal liver-derived RBCs like CB cells are described to have a higher volume and Hb content in comparison to BM-derived adult RBCs [10,11]. Interestingly, the Hb concentration in adult nRBCs was lower than in cRBC_iPSC and CB-derived nRETs, but comparable to cRBCs derived from adult HSPCs (cRBC_adult) [10]. This might indicate a developmental impact on the Hb content rather than a culture-related phenomenon. Although the ex vivo expansion of cRBC_adult is out of the focus of the present paper, for comparison reasons, data obtained from cRBC_adult expanded in the same culture system described here for cRBC_iPSCs, are summarized in Appendix A. Finally, SEM verified the morphology of cRBC_iPSCs comparable with that of nRETs or even pro-biconcave structured RBCs. Most importantly, no prominent characteristics indicating a defective membrane organization like echinocyte formation were noticeable.

The two main biophysical membrane characteristics of RBCs are deformability and OR. Our results verified higher OR in nRETs than in nRBCs, as previously reported [10,34]. OR of cRBC_iPSCs was comparable with that of nRETs. OR is affected by the membrane surface area-to-volume ratio (S/V ratio) and the membrane integrity [35,36]. The results would therefore argue for membrane S/V ratio and membrane integrity of cRBC_iPSCs comparable with that of nRETs. It is well established that young reticulocytes exhibit declined membrane deformability, as also seen in this study [28,29,37]. However, the deformability of cRBC_iPSCs was somewhat lower than that of nRETs and varied significantly from that of nRBCs. To the best of our knowledge, thus far only Kobari et al. examined the deformability of cRBC_iPSCs obtained from a patient with sickle cell anemia and reported similarity to healthy controls [38]. Nevertheless, in that study, deformability was induced by osmotic alterations rather than shear stress and a wide range for healthy controls was given (EImax 0.38–0.58) in comparison with the present study (EImax 0.57–0.61). Deformation capacity of human RBCs is a function of (1) geometry of the cells, especially the membrane S/V ratio; (2) intracellular viscosity determined by the Hb content and the hydration state; and (3) membrane composition [26,39]. In cRBC_iPSCs, these features may vary from native cells because of the culturing process or the more primitive nature of iPSC-derived cells. Larger cell size and a higher Hb content that might affect the S/V ratio have also been described for cRBC_adult and were ascribed to stress erythropoiesis and enhanced fetal hemoglobin (HbF) expression [31,40,41,42]. However, the deformability of cRBC_adult was reported to be comparable with reticulocytes [10,40,43]. These observations may contend for a developmental influence on cRBC_iPSCs deformability rather than culture-related artefacts.

During human development, hematopoietic cells arise in overlapping waves. Primitive erythroblasts in the yolk sac express the embryonic globin genes Gower I and Gower II. Cells derived from fetal liver erythropoiesis, like CB-derived nRET, express primarily HbF. After birth, hematopoiesis switches from the fetal liver into the bone marrow (BM). BM-derived RBCs express mainly adult hemoglobin (HbA) [12]. In our system, cRBC_iPSCs are majorly HbF positive, with negligible contributions by HbA, but up to 18% embryonic Hb, as already published [9]. The detection of embryonic Hb specifies at least a partial contribution of primitive RBCs to the examined cRBC_iPSC population, whereas the majority of cells seems to represent the fetal phenotype [12]. Whereas definitive RBCs mature mainly extravascular in secured BM niches, primitive RBCs enucleate and mature in the blood stream [12,44]. They are up to 80% larger than cells from the definitive lineage [45]. Besides the expression of embryonic and fetal Hb, enhanced cell size, volume, and Hb content of cRBC_iPSCs in our system are consistent with a more primitive nature of cells. These parameters might affect the S/V ratio of cells and, thus, their deformability. Variance in viscosity between embryonic Hb, HbF, and HbA might further affect cell rheology. In line with our observations, Olivier reported a 15% larger cell size of iPSCs-derived cRBCs than of cRBC_adult [46]. Knowledge regarding the membrane composition and biomechanical characteristics of primitive RBCs is poor. Recent investigations in mice suggested similar expression of major membrane skeleton genes in primitive and definitive erythroblasts [44,47]. Investigations on primitive RBCs of human origin are still missing. One proteomics analysis of primitive human RBCs demonstrated the occurrence of numerous cytoskeletal proteins, including band 3, and ankyrin, which are also of relevance in definitive cells [48].

The bending/elasticity of the membrane further depends on its well-balanced composition. The lipid bilayer of the RBC membrane comprises equal amounts of cholesterol (chol) and phospholipids [26]. We already conducted lipidomics analyses of nRETs and nRBCs. In line with their larger cell surface area, nRETs exhibited a higher absolute amount of lipids but a comparable pattern in proportions of individual lipid classes [10]. These results were verified in this study. Main membrane lipids were chol, followed by the phospholipids PC, SM, and PE. The same lipid pattern was found in cRBC_iPSCs. Like in nRETs, the absolute contents of chol and each discrete phospholipid class (PS, PC, PI, PE, and SM) were more abundant in cRBC_iPSCs than in nRBCs. The chol content significantly affects the OR of cRBCs, and appears to be reduced in ex vivo cultures due to malnutrition of cells [10]. In the present study, chol was added during culturing, resulting in a chol content of cRBC_iPSCs (45%) near values noticed in nRETs and nRBCs (50%). This was adequate to rescue cRBC_iPSCs from enhanced osmotic fragility and to obtain comparability with nRETs in OR and morphology. Interestingly, LPC was the only phospholipid that was lower in nRETs and cRBC_iPSCs than in nRBCs and appeared to be enriched during membrane reorganization to biconcavity. LPC, also known as lysolecithin, has been applied in numerous studies to induce RBC membrane vesiculation [39,49]. Because of its single hydrocarbon chain, LPC could generate packing defects in the membrane interior, thereby reducing the bilayer rigidity. Minor alterations in the LPC content might be of relevance for terminal membrane properties.

The lipid bilayer is linked to the cytoskeletal spectrin-aktin network through interactions based on two macromolecular complexes, one ankyrin based and the other 4.1R based [26]. IF patterns for the cytoskeletal protein spectrin and the adaptor protein ankyrin, involved in vertical linkage between the membrane proteins and the cytoskeleton, were comparable between cRBC_iPSCs and the native counterparts. Both, the ankyrin and the 4.1R complex constitute the anion transporter band 3, the most abundant membrane protein. A decline in band 3 causes malfunction and release of vesicles [50,51]. We did not see obvious signs for band 3 alterations in cRBC_iPSCs. F-actin, together with tubulin and myosin IIβ, are essential in enucleation, forming the enucleosome [28,52]. Later in erythroid maturation, cytosolic actin and tubulin are no longer critical and are removed. Accordingly, in our study immature cRBC_iPSCs and nRETs showed local accumulations of f-actin, supposed to be remnants of the contractile actin ring established during enucleation.

Culturing RBCs from an unrestricted source, such as iPSCs (homozygous for transfusion relevant blood group antigens or expressing rare antigens), is a promising option for donor independent production of RBC test panels. To validate the suitability of cRBC_iPSCs for these diagnostic tasks, we analyzed the expression of clinically most important blood group antigens. The expression of ABO, Rhesus, Kell, Kidd, MN, and Ss antigens were similar to the genotype of undifferentiated iPSCs. These outcomes demonstrated that cRBC_iPSCs would already be compatible with diagnostic applications. However, the detection of the Duffy (Fy) antigen Fyb failed in cRBC_iPSCs. This was independent of its homozygous or heterozygous expression. Genetic variants that cause weak or null expression of Fyb were ruled out by genotyping. Conversely, the expression of Fya was effortlessly obvious. The Duffy glycoprotein is a transmembrane protein that spans the RBC membrane seven times. The antigen has previously been known as Duffy Antigen Receptor for Chemokines (DARC), encoded by the FY gene on chromosome 1. The two main codominant alleles, *FY*01 (FY*A*) and *FY*02 (FY*B)* differ by a single nucleotide at position 125 (G versus A) and encode Fya and Fyb antigens that differ by a single amino acid at residue 42 (glycine versus aspartic acid) [53,54]. Interestingly, Fyb antigen expression was detectable in cRBCs derived from adult HSPCs by using the same erythroid differentiation protocol (data not shown). Therefore, the iPSC origin of cRBCs appears to be of significance for this disparity rather than the culture system. Thus, we hypothesize that altered gene methylation, post-transcriptional protein modification, or degradation of the Fyb antigen appears to occur. At this state, the cause of this failure is uncertain and requires more in-depth analyses.

In summary, cRBC_iPSCs generated in a cell culture system providing cellular interactions and optimized lipid supplementation exhibited comparable morphology and membrane characteristics with nRETs derived from CB, but less deformability. Reduced deformability might reflect the membrane features of primitive erythroid cells, although this remains speculative since we cannot completely exclude an altered cellular structure due to the in vitro culturing process. Especially the incomplete simulation of the physiological microenvironment might be of relevance. Cellular interactions provided by the bone marrow, the spleen, the vascular endothelium, or even the shear stress in the blood flow could be significant for adequate and complete membrane remodeling, as recently shown by further maturation of iPSC-derived cRBCs after transfusion into mice [38,55]. Although numerous challenges have to be overcome before cRBC_iPSCs gain entrance into clinical applications, RBC membrane properties obtained in this study underscore the overall prospects of cRBC_iPSCs for their future application in transfusion medicine. As an example, initial blood group phenotyping of cRBC_iPSCs gave promising results for their application as RBC test panels in the near future.

## Figures and Tables

**Figure 1 cells-11-02473-f001:**
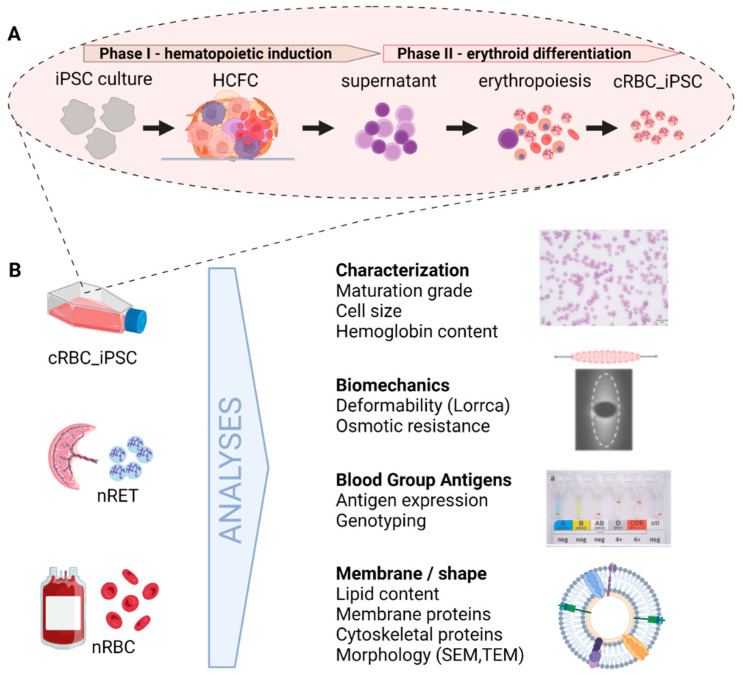
Schematic presentation of the study design. (**A**) Generation of cRBC_iPSCs in an established culture system for ex vivo production of cRBC from human iPSCs [9]. First, undifferentiated iPSCs formed self-organized three-dimensional plastic-adherent cell aggregates, termed “hematopoietic cell forming complex” (HCFC), which were stimulated with stem cell factor (SCF), EPO, and IL-3 (phase I, hematopoietic induction). After 3–5 weeks, hematopoietic cells were released into the supernatant from these HCFCs and harvested for further erythroid differentiation for an additional 18 days (phase II, erythroid differentiation). Finally, erythroid cells were filtered to obtain the pure enucleated proportion of cRBC_iPSCs. (**B**) Enucleated cRBC_iPSCs were characterized for their phenotype and maturation stage and compared with nRETs and nRBCs for morphology, biomechanical characteristics, membrane composition, and blood group antigen expression.

**Figure 2 cells-11-02473-f002:**
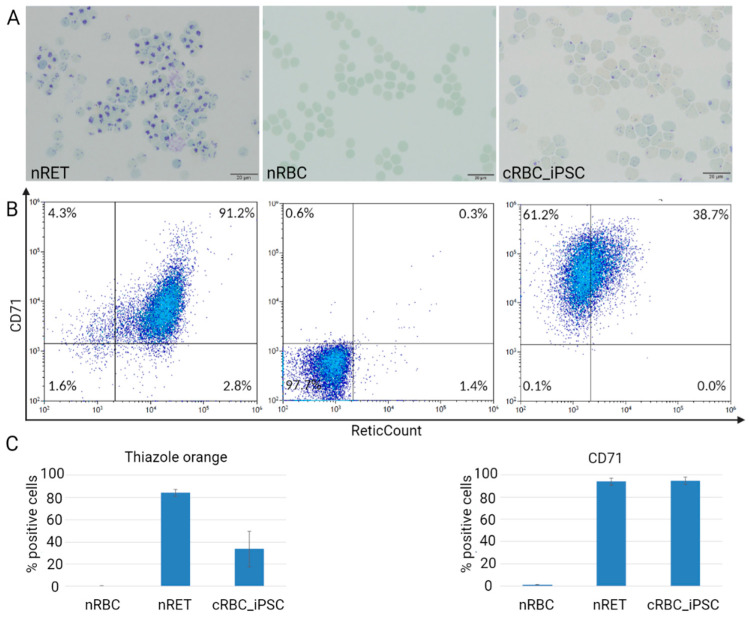
Maturation stage of cRBC_iPSCs compared with those of nRETs and nRBCs. (**A**) Representative cytospin preparations stained with new methylene blue for the residual nucleic acid (shown are cRBCs derived from iPSC3; cRBCs derived from the other iPSC lines are shown in Appendix A). (Scale bar = 20 µm, magnification of 100× with oil). (**B**) Representative flow cytometry analyses for thiazole orange and CD71 (shown are cRBCs derived from iPSC4; cRBCs derived from the other iPSC lines are shown in Appendix A). (**C**) Percentages of thiazole orange and CD71 positive cells determined by flow cytometry, nRETs (n = 5), nRBCs (n = 4), and cRBC_iPSCs (n = 16) (mean ± SD).

**Figure 3 cells-11-02473-f003:**
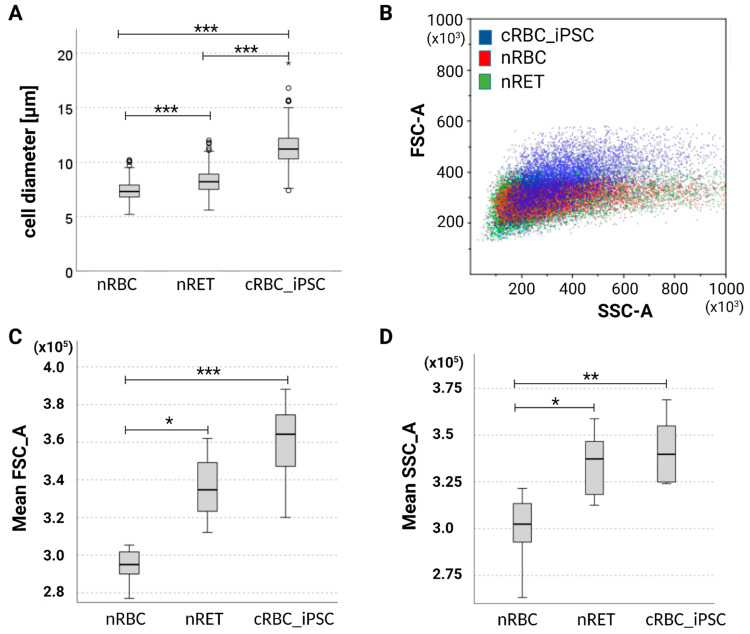
Cell size of cRBC_iPSCs compared with those of nRBCs and nRETs. (**A**) Differences in cell diameter determined by microscopy (cytospin slides from four different iPSC lines were enumerated and n = 300 cells were counted per slide as visualized in Appendix A). (**B**) Representative forward (FSC-A) and side scatter (SSC-A) properties of cells determined by flow cytometry with identical instrument settings. (**C**) Box and whisker plots depicting variance in FSC-A and (**D**) SSC-A characteristics (mean fluorescence intensity) of cRBC_iPSCs (n = 7), nRBCs (n = 9), and nRETs (n = 8). Significance is given as * *p* < 0.05; ** *p* < 0.01; *** *p* < 0.001.

**Figure 4 cells-11-02473-f004:**
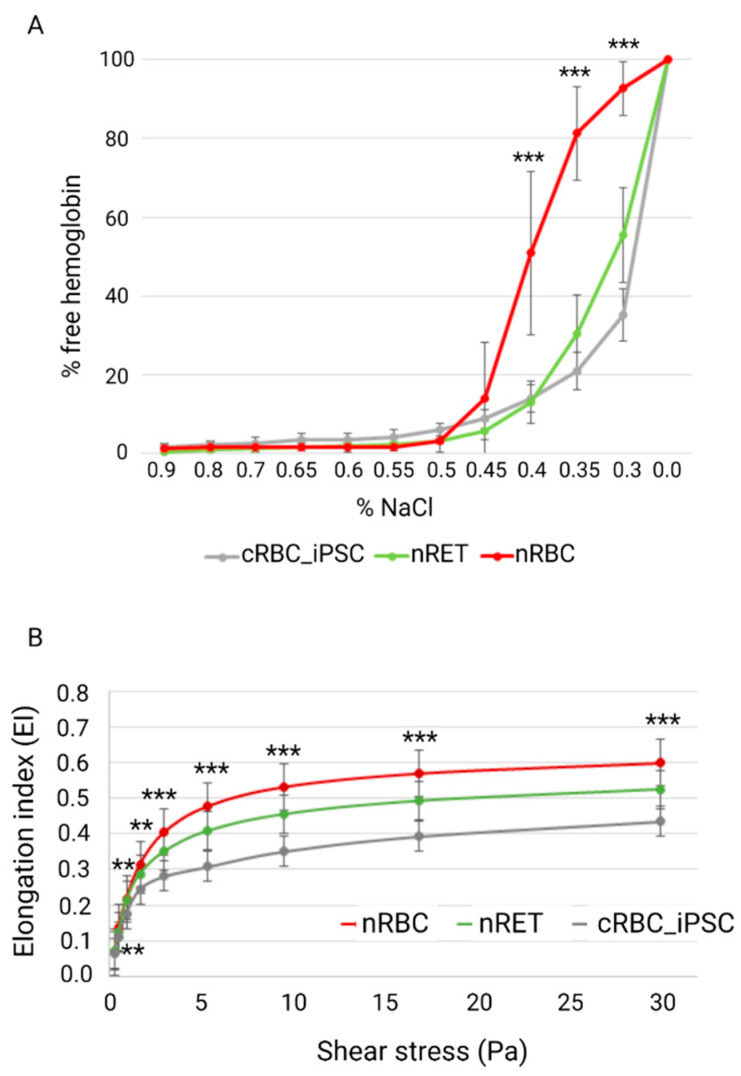
Biomechanical properties of cRBC_iPSCs, nRETs, and nRBCs. (**A**) Osmotic resistance of cRBC_iPSCs (n = 8), nRETs (n = 4), and nRBCs (n = 14) evaluated by the amount of free Hb under declining NaCl concentrations. Although the cRBC_iPSCs and nRETs showed a comparable OR, significant differences between cRBC_iPSCs and nRBCs were seen in the range 0.4% to 0.3% NaCl. (**B**) Deformability measured by Lorrca analysis comparing nRETs (n = 4), nRBCs (n = 13), and cRBC_iPSCs (n = 8). Lower deformability of cRBC_iPSCs reached significance only in comparison with nRBCs (** *p* < 0.01, *** *p* < 0.001).

**Figure 5 cells-11-02473-f005:**
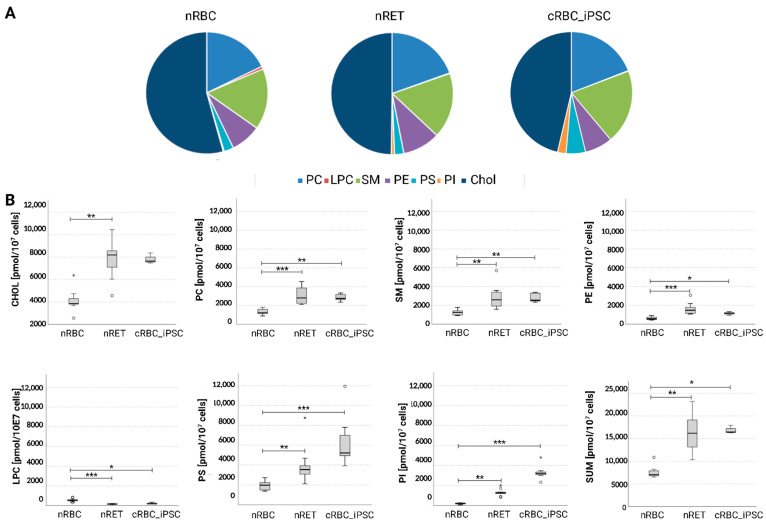
Total cellular lipid content analyzed via high-resolution mass spectrometry. (**A**) Pie charts displaying the proportional (%) content of PE, PC, PS, PI, SM, LPC, and chol of cRBC_iPSCs (n = 8), nRBCs (n = 9), and nRETs (n = 9). (**B**) Absolute content of PE, PC, PS, PI, SM, LPC and chol, represented as pmol/10^7^ cells, compared between cRBC_iPSCs, nRBCs, and nRETs. * *p* < 0.05, ** *p* < 0.01, *** *p* < 0.001. ° statistical outliers.

**Figure 6 cells-11-02473-f006:**
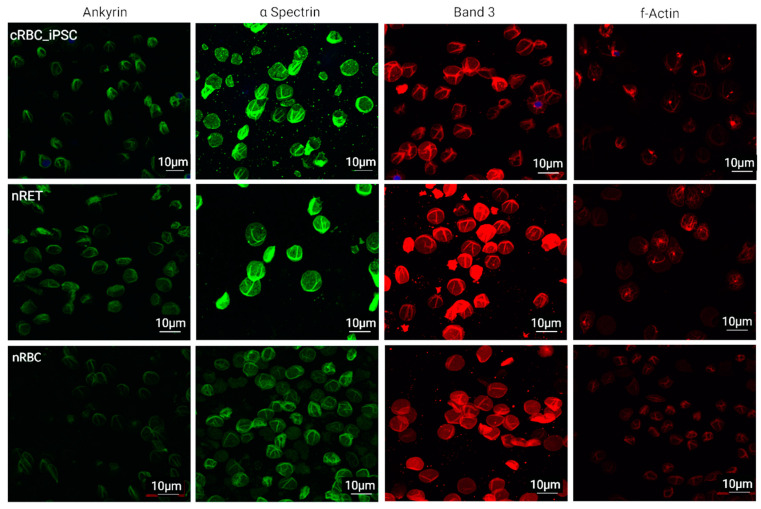
Immunofluorescence analyses of membrane and skeletal proteins. Representative immunofluorescence staining of the cytoskeletal proteins ankyrin, α-spectrin, f-actin, and membrane protein band 3 in cRBC_iPSCs, nRETs, and nRBCs (Scale bar = 10 µm, magnification of 60× with oil). No differences in the MFI could be observed between the different sources (Appendix A).

**Figure 7 cells-11-02473-f007:**
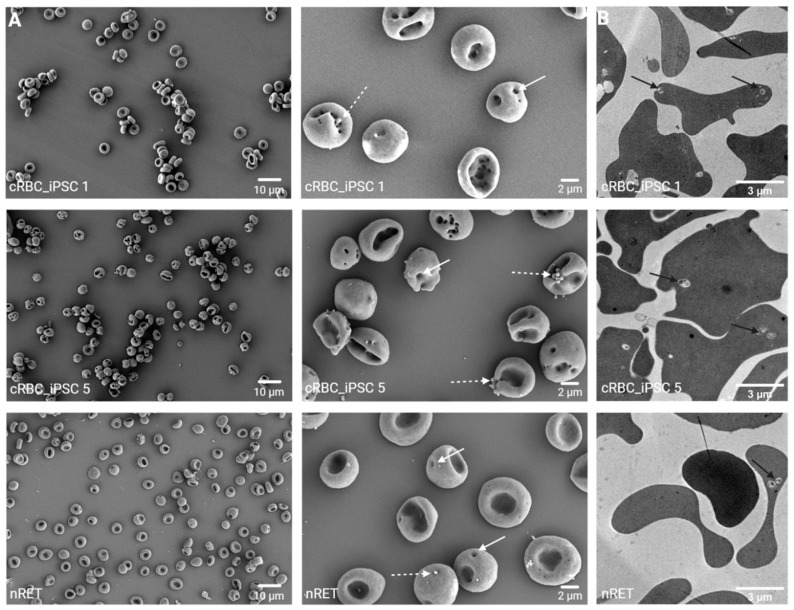
Electron microscopy analysis of cRBC_iPSCs. (**A**) Representative SEM pictures of enucleated cRBC_iPSC 1 and cRBC_iPSC 5 and nRET (scale bars: 10 and 2 µm). In higher magnification pictures, pit-like structures (continuous arrows) and bubble-like excesses (dotted arrows) in the membrane of cRBC_iPSCs and nRETs are noticeable. (**B**) Representative TEM pictures. Black arrows mark large intracellular vesicles, comparable with MVE, containing residual degraded cell organelles.

**Figure 8 cells-11-02473-f008:**
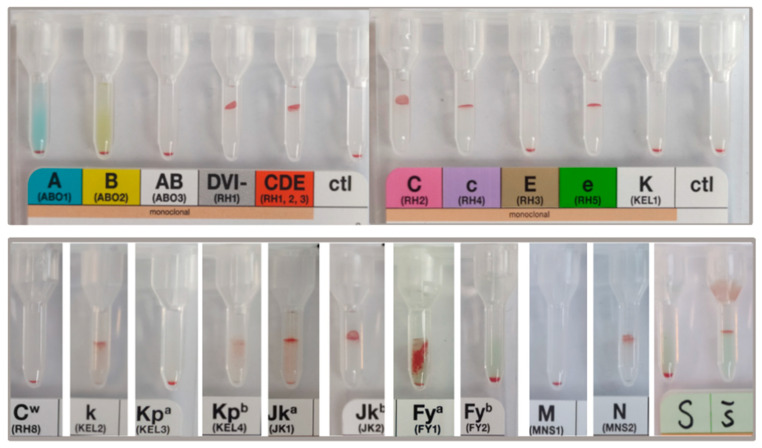
Expression of blood group antigens by cRBC_iPSCs 1 (phenotyping). Expression of ABO, Rhesus, Kell, Kidd, Duffy, and MNSs antigens were evaluated using commercial accessible test systems for visual evaluation in gel cards. The cRBC_iPSCs shown here are O Rh (D) pos, CcD.ee, Cw-, Kell-k+, Kpa-Kpb+, Jka+Jkb+, Fya+Fyb-, M-N+, S-s+. Controls (ctl) are negative, thus, validating the obtained results. Appendix A show phenotyping of cRBC_iPSCs 1 and cRBC_ iPSCs 5.

**Table 1 cells-11-02473-t001:** Blood group antigen expression of cRBC_iPSCs. (**A**) Phenotype of cRBC_iPSCs and (**B**) genotype of corresponding iPSC lines. The blood group antigens determined are listed in the table. The nomenclature according to the International Society of Blood Transfusion (ISBT) is used (isbtweb.org). * Blood group allele terminology: Alleles are designated by the system symbol, followed by an asterisk, and antigen number. Discrepant results of antigen phenotyping and genotyping are colored in red.

**A**	**Blood group phenotypes cRBC_iPSCs**					
**Samples**	**ABO**	**Rhesus**	**Kell**	**Kidd**	**Duffy**	**MNS**
cRBC_iPSC1	O	D, c, E, e, C^w^−	K−, k+, Kp(a−b+)	Jk(a+b−)	Fy(a−b−)	M+N+S+s+
cRBC_iPSC3	O	D, C, c, e, C^w^−	K−, k+, Kp(a−b+)	Jk(a+b+)	Fy(a+b−)	M−N+S−s+
cRBC_iPSC5	O	D, C, c, e, C^w^−	K−, k+, Kp(a-b+)	Jk(a+b+)	Fy(a+b−)	M+N−S−s+
**B**	**Blood group genotypes undifferentiated iPSCs**					
**Samples**	**ABO**	**Rhesus**	**Kell**	**Kidd**	**Duffy**	**MNS**
iPSC1	ABO*O.01.01/ABO*O.01.02	ccD.Ee	KEL*02, KEL*02.04	JK*01	FY*02	GYPA*01/GYPA*02 GYPB*03/GYPB*04
iPSC3	ABO*O.01.01/ABO*O.01.01	CcD.ee	KEL*02, KEL*02.04	JK*01/JK*02	FY*01/FY*02	GYPA*02/GYPA*04
iPSC5	ABO*O.01.01/ABO*O.01.02	CcD.ee	KEL*02 KEL*02.04	JK*01/JK*02	FY*01/FY*02	GYPA*01/GYPA*04

## Data Availability

Mass spectrometry data are available upon request to the corresponding author.

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
