# Peer review of "Membrane Properties of Human Induced Pluripotent Stem Cell-Derived Cultured Red Blood Cells"

_cells, 2022, doi:10.3390/cells11162473_

Round 1

Reviewer 1 Report

Bernecker et al. reported the membrane properties of human induced pluripotent stem cell-derived cultured red blood cells. This study established culture systems for cultured red blood cells from human induced pluripotent stem cells (cRBC_iPSCs). They studied the morphology and maturation stage of cRBC_iPSCs were closer to native reticulocytes (nRETs) and osmotic resistance of cRBC_iPSCs was similar to nRETs. This is an interesting study but not enough novelty to publish in Cells. 

Author Response

Dear Editor 1,

thank you very much for your comments. Based on comments from reviewer 2 and 3, we tried to improve the quality of our manuscript. From our point of view, data describing the characteristics of cRBCs from human iPSCs are poor, and a comprehensive study that includes features like osmotic resistance, deformability and lipid analyses are still missing. The main limitation is still the insufficient generation of enucleated RBCs from human iPSCs. However, we agree that additional work will be necessary to completely analyze and finally improve the membrane characteristics of cultured RBCs from human iPSCs for their future clinical or diagnostical application. 

Kind regards, Isabel Dorn 

Reviewer 2 Report

Bernecker and colleagues are exploring the remarkable membrane properties of cRBC_iPSC. The authors provide compelling evidence for the comparison of various membrane properties such as membrane morphology, lipid composition and membrane-associated proteins with nRETS and nRBCs. The authors also checked the blood group antigen expression. Overall, the data support the conclusions very well and the study is well designed and executed. 

Bernecker and colleagues are exploring the remarkable membrane properties of human-induced pluripotent stem cells (cRBC_iPSC), native reticulocytes (nRETs) and native red blood cells (nRBCs). Firstly, author compared the maturation stages of cRBCs_iPSCs with nRETs and nRBCs. Later they checked the biomechanical properties of RBCs such as osmatic pressure and deformability. The authors also measured and compared the membrane lipid composition between cRBCs_iPSCs, nRETs and nRBCs by high-resolution mass spectrometry. Furter authors also checked and compared the expression of cytoskeletal and membrane protein and observed the morphology of cRBC_iPScs by electron microscopy. The authors also analyzed the expression of blood group antigens. Overall, the data support the conclusions very well and the study is well designed and executed, and I have only a few suggestions for improving the manuscript.

1.  Line 214: Authors state that degradation of intracellular components…… It is not clear what does author means by this statement. Also, what intracellular components are degraded upon maturation? Authors could cite an appropriate reference.

2. Fig3 A: How did the authors measure the cell diameter? It would be appropriate to include some microscopy images to analyze.

3. Figure 5: Does author measure the total lipid composition of the cells or authors extract the plasma membrane and measured the lipid composition. It’s not clear from the method and result section. It would be critical to determine the integrity/purity of the membrane fraction to determine the membrane composition.

4. Figure 6: It would be appropriate to quantify the total fluorescence and plot the comparison. There is slight less fluorescence intensity of Ankyrin in nRBC.

Author Response

Dear Reviewer 2,

thank you very much for your constructive criticisms. Based on your comments, we performed the following changes / additions:

Comment 1: This sentence (now 224-225) was modified for better understanding and two references were added (22, 23).

Comment 2: We included a new Supplementary Figure S4 in order to show representative microscopy images obtained from cell diameter measurements, as indicated in lines 204-241 and in the Figure legend of Figure 3. 

Comment 3: In our study, we measured the total lipid composition of the cells. We clarified this in lines 171-172, 291, and in the Figure legend of Figure 5. In contrast to other cell types like hepatocytes, the intracellular lipid content of RBCs seems to be very low and analyses might be more biased by the plasma membrane extraction process than by the low intracellular lipid content.

Comment 4: Thank you very much for this suggestion. The mean fluorescence over the image fields was calculated and results were summarized in the new Supplementary Figure S5 (as well as in the result section lines 318-320, Figure legend of Figure 6, and in the Material and Methods part, lines 164 - 166). Despite the slight less fluorescence of ankyrin in Figure 6, no statistical differences were observed in MFI of Ankyrin between the different sources.

We hope that the modified version of our manuscript is now appropiate for publication in CELLS.

Kind regards, Isabel Dorn

Reviewer 3 Report

The authors of this manuscript provide a thorough and detailed characterization of all relevant parameters of iPSC-derived enucleated cRBC in comparison with native erythrocytes and reticulocytes.

There are a few points of attention.

The comparison is made with native ery’s derived from adult blood, and native reticulocytes derived from cord blood. I miss a discussion on the difference that can be expected between these two controls because of the different developmental stage. What are the differences between native ery’s from adult blood and cord blood. These differences are indicative of differences due to developmental stage versus differences due to differentiation stage.

Similarly, it would have been interesting to know how cRBC cultured from iPSC compare to cRBC cultured from adult blood. It is not realistic to perform all experiments again, but literature may give some indications?

The expression level of CD71 remained high. The authors may want to discuss whether there may not be sufficient iron (holotransferrin) in the medium. The authors use 330 mg/L and maximum proliferation may require 1g/L?  Lack of iron increases CD71 expression.

This may be related to hemoglobin levels. The title of paragraph 3.2 includes hemoglobin but only the last sentence mentions similar Hb concentrations in nRET and iPSC-cRBC with a value of ca. 205 mg/ml. I suppose this is the MCHC? Can the authors specify? In that case the control level should be above 300 mg/ml?? What is the level in nRBC ??

Author Response

Dear Reviewer 3,

thank you very much for your constructive comments. Based on your suggestions, we performed the following changes and additions in our manuscript:

Comment 1: In the revised version we tried to explain and highlight the developmental differences between cord-blood derived cells and peripheral blood-derived cells in more detail (Introduction lines 80-86, Discussion 404-407).

Comment 2: We already performed most of the experiments also with cRBCs derived from adult HSPCs and included these data in the „initial draft“ of our manuscript. However, we thought that the manuscript was very difficult to read and to understand with all the data from different sources. Therefore, we removed this part for better understanding. Based on your comment, we now included data from cRBCs derived from adult HSPCs for comparison reasons in Supplementary table S3, and discussed these data in lines 409-415.

Comment 3 - prolonged CD71 expression: Thank you very much for this comment. We will immediately perform some experiments to test for the hypothesis that insufficient iron supply during large scale amplification might be the reason for prolonged CD71 expression in our experiments. A comment was added in the discussion section (lines 399-402).

Comment 4 - Hemoglobin content measured by Drabkin‘s solution: As suggested, we added the obtained values for adult nRBCs in lines 251-254. Due to differences in measurement technologies, we do not believe that the Hb concentration obtained by Drabkin‘s staining is comparable with the MCHC. Interestingly, the Hb concentration in adult nRBCs was lower than in cRBC_iPSC and CB-derived nRETs, but comparable to cRBCs derived from adult cRBCs derived from HSCs in the same erythroid ex vivo culture system (see new Supplementary Table S3). This might indicate for an developmental impact on the Hb content of cRBC_iPSCs rather than a culture phenomenon as now discussed in lines 409 – 415 and lines 453-455.

We hope that the modified version of our manuscript is now appropiate for publication in CELLS.

Round 2

Reviewer 1 Report

The author improved the mansucript based on the other two reviewer's comments.